# Selective Sampling-based Scalable Sparse Subspace Clustering

**Shin Matsushima**
University of Tokyo
smatsus@graco.c.u-tokyo.ac.jp

**Maria Brbić**
Stanford University
mbrbic@cs.stanford.edu

## Abstract

Sparse subspace clustering (SSC) represents each data point as a sparse linear combination of other data points in the dataset. In the representation learning step SSC finds a lower dimensional representation of data points, while in the spectral clustering step data points are clustered according to the underlying subspaces. However, both steps suffer from high computational and memory complexity, preventing the application of SSC to large-scale datasets. To overcome this limitation, we introduce Selective Sampling-based Scalable Sparse Subspace Clustering ($S^5C$) algorithm which selects subsamples based on the approximated subgradients and linearly scales with the number of data points in terms of time and memory requirements. Along with the computational advantages, we derive theoretical guarantees for the correctness of $S^5C$. Our theoretical result presents novel contribution for SSC in the case of limited number of subsamples. Extensive experimental results demonstrate effectiveness of our approach.

## 1 Introduction

Subspace clustering algorithms rely on the assumption that high-dimensional data points can be well represented as lying in the union of low-dimensional subspaces. Based on this assumption, the task of subspace clustering is to identify the subspaces and assign data points according to the corresponding subspaces [1]. The clustering task is usually performed in two steps: (i) representation learning; and (ii) spectral clustering. In the representation learning step the goal is to find a representation of data points according to the underlying low-dimensional subspaces. The obtained representation is then used to construct the affinity matrix whose entries define similarity between data points. Ideally, the affinity matrix is block diagonal and non-zero values are assigned only to data points lying in the same subspace. Given an affinity matrix as an input, spectral clustering [2] assigns subspace membership to data points. In particular, spectral clustering defines clustering problem as a minimum cut problem on a graph and minimizes relaxed versions of the originally NP-hard normalized cut (NCut) [3, 4] or ratio cut (RCut) [5] objective functions.

Subspace clustering algorithms often differ in regularizations imposed on the representation matrix, such as sparsity [6–8], low-rankness [9, 10], or their combination [11–13]. In this paper we are interested in the Sparse Subspace Clustering (SSC), proposed by Elhamifar and Vidal [7]. SSC imposes sparsity constraint on data representation matrix by solving the $\ell_1$ norm regularized objective. SSC enjoys strong theoretical guarantees and can succeed in the noiseless case even when subspaces intersect [14]. Moreover, SSC is provably effective with noisy data as long as the magnitude of noise does not exceed a certain threshold [15]. Tsakiris and Vidal [16] recently established guarantees for SSC with missing data.

Despite the strong theoretical guarantees [14, 15] and superior performance [7], a key challenge towards the wide applicability of SSC lies in the development of methods able to handle large-scale data. In particular, learning representation matrix takes $\mathcal{O}(N^3)$ operations in ADMM-based solver

used in SSC, where $N$ is the number of data points. Pourkamali-Anaraki and Becker [17] address this problem by proposing more efficient implementation based on the matrix-inversion lemma; however, the method still requires $\mathcal{O}(N^2)$ operations. The same problem is present in the spectral clustering step which performs eigenvalue decomposition of the Laplacian matrix, resulting in polynomial time complexity. In addition to high time complexity, the memory cost of SSC requires $\mathcal{O}(N^2)$ space. Overall, high time and space complexity limit the application of SSC to small or moderately sized datasets. Since unlabeled data is often easily obtainable, this limitation is in contrast with many real-world clustering tasks.

Motivated by the above challenges, we propose Selective Sampling-based Scalable Sparse Subspace Clustering ($S^5C$) algorithm which linearly scales with the number of data points in terms of computational and memory requirements. Instead of relying on a random subsample [18–20], the key idea of our approach is to select data points in terms of the most violating subgradient of the objective function. In the representation learning step, we solve a small number of LASSO problems [21] and select subsamples in an iterative manner. Once representation matrix is obtained, we perform spectral clustering by approximating eigenvectors of graph Laplacian using the block version of the power method. Whereas in general setting power method suffers from quadratic complexity, $S^5C$ achieves linear time and space by the guarantee to have at most $O(N)$ elements different from zero in the subspace learning step.

From the theoretical aspect, we provide approximation guarantees under which subspace detection property of $S^5C$ is preserved. Our main result states that SSC can exactly recover subspaces even in the case of limited number of subsamples, where the number of subsamples is independent of data size. This notable result has a broader significance and can be applied to other sampling-based SSC algorithms, such as [19, 20]. Compared to random sampling, theory implies selective sampling is advantageous. Extensive experiments on six real-world datasets of varying size demonstrate the superior clustering performance of $S^5C$ compared to the state-of-art large-scale sparse subspace clustering algorithms. Considering that all existing methods avoid to directly solve $\ell_1$ regularized basis pursuit problem, to the best of our knowledge, this is the first method with the original SSC formulation that scales linearly with the number of data points.

## 1.1 Related work

**Algorithmic aspect.** Much of the existing work has been devoted to scaling representation learning step in SSC. Although more efficient than SSC, Orthogonal Matching Pursuit (OMP) [22, 23] and nearest neighbor based SSC methods [24, 25] do not scale well to large datasets. Scalable Sparse Subspace Clustering (SSSC) [19, 20] randomly samples small set of data points and performs SSC. Out-of-sample data points are then classified by minimizing the residual over the in-sample data. Although this method solves large $N$ problem, the original SSC is still performed on a small-scale dataset. Furthermore, relying only on a random subsample can result in weak performance in the cases when the subsample is not representative of the original dataset. All existing methods avoid to directly solve $\ell_1$ regularized basis pursuit problem. In contrast to them, $S^5C$ preserves the original construction of the affinity matrix of SSC. In the spectral clustering step, most existing methods apply computationally inefficient spectral clustering. Power Iteration Clustering (PIC) [26] has been proposed as a fast and scalable alternative to spectral clustering. However, if PIC is applied to SSC, theoretical guarantees of SSC do not hold anymore. On the other hand, our spectral clustering step preserves theoretical guarantees, while retaining advantages of PIC: fast convergence, scalability and simple implementation. Furthermore, experiments show it achieves significantly better performance than PIC.

**Theoretical aspect.** Another limitation of the existing work lies in inability to preserve desirable theoretical properties of SSC. EnSC-ORGEN [27] and SSC-OMP [23] derive scalable active set method and prove subspace preserving property for arbitrary subspaces. However, their guarantee holds only in a finite number of subsamples which can be all data points, and therefore, does not ensure that the algorithm is more efficient than SSC. Recently proposed exemplar-based subspace clustering [28] selects subset of data points such that robustness to imbalanced data is achieved and constructs affinity matrix by nearest neighbor. Although it has linear time and memory complexity, it fails to prove subspace preserving property except in the setting of independent subspaces which is overly restrictive assumption [29]. SSSC [19, 20] relies on a random subset selection and does not provide any theoretical justification. Whereas our focus in this work is on selecting samples based on

subgradient approximation instead of random sampling, we show how our theoretical results can be readily extended to random sampling case. Table 1 summarizes relation of our theoretical analyses to the analyses of existing work.

Table 1: Relation to the existing theoretical results

| | Subsample | Noise | Data model | Measure for subspaces | Condition on data |
|---|---|---|---|---|---|
| Theorem 2 in [23] | no | no | deterministic | incoherence | large inradius |
| Theorem 2.8 in [14] | no | yes | semi-random | affinity | large number of data |
| S$^5$C Theorem 1 | yes | no | deterministic | incoherence | large persistent inradius |
| S$^5$C Theorem 2 | yes | no | semi-random | affinity | large number of data |

## 2    Sparse subspace clustering

Consider data matrix $\mathbf{X} \in \mathbb{R}^{M \times N}$ whose columns are $N$ data points drawn from a union of $L$ linear subspaces $\bigcup_{\ell \in [L]} \mathcal{S}_\ell$ of unknown dimensions $\{d_\ell\}_{\ell \in [L]}$ in $\mathbb{R}^M$. Sparse subspace clustering (SSC) solves the following optimization problem:

$$\operatorname*{minimize}_{\mathbf{C} \in \mathbb{R}^{N \times N}} \quad \frac{1}{2} \|\mathbf{X} - \mathbf{X}\mathbf{C}\|_F^2 + \lambda \|\mathbf{C}\|_1, \text{ subject to } \operatorname{diag}(\mathbf{C}) = \mathbf{0}, \tag{1}$$

where $\mathbf{C} \in \mathbb{R}^{N \times N}$ is representation matrix and $\lambda$ is a hyperparameter for sparsity regularization. SSC solves the resulting convex optimization problem using the ADMM solver [30, 7]. Once representation matrix is obtained, affinity matrix $\mathbf{W} \in \mathbb{R}^{N \times N}$ is constructed to achieve symmetry as $\mathbf{W} = |\mathbf{C}| + |\mathbf{C}|^\top$.

Given affinity matrix $\mathbf{W}$ and number of clusters $L$, SSC applies spectral clustering algorithm [4, 2]. Specifically, it finds $L$ eigenvectors corresponding to the $L$ smallest eigenvalues of the symmetric normalized graph Laplacian matrix defined as $\mathbf{L}_S = \mathbf{I}_N - \mathbf{D}^{-\frac{1}{2}} \mathbf{W} \mathbf{D}^{-\frac{1}{2}}$, where $\mathbf{D} \in \mathbb{R}^{N \times N}$ is diagonal degree matrix in which $(i, i)$-th element is the sum of $i$-th column of $\mathbf{W}$. Given matrix whose columns are $L$ eigenvectors, cluster memberships of data points are obtained by applying $K$-means algorithm to the normalized rows of the matrix.

## 3    Selective sampling-based SSC

In this section, we first propose how to efficiently learn representation matrix in SSC, and then propose the solution for scaling spectral clustering step. Time and memory complexity of S$^5$C algorithm are analyzed in Appendix A.

### 3.1    Representation learning

In the representation learning step we aim to solve SSC problem in (1) using only a small number of selectively sampled data points instead of the entire data matrix $\mathbf{X}$. Let $C_{ji}$ denote $(j, i)$-th element of $\mathbf{C}$ and $\mathbf{x}_i \in \mathbb{R}^M$ denote $i$-th column of $\mathbf{X}$. The problem in (1) can be decomposed by $N$ problems, where the following problem needs to be solved for $i$-th column of $\mathbf{C}$:

$$\operatorname*{minimize}_{(C_{ji})_{j \in [N]} \in \mathbb{R}^N} \frac{1}{2} \left\| \mathbf{x}_i - \sum_{j \in [N]} C_{ji} \mathbf{x}_j \right\|_2^2 + \lambda \sum_{j \in [N]} |C_{ji}|, \text{ subject to } C_{ii} = 0. \tag{2}$$

Note that for each $i \in [N]$ the decomposed problem in (2) has $O(N)$ parameters, so the resulting time and space complexity is $O(N^2)$ which is not acceptable for large-scale data.

Following the basic subspace clustering assumption that data points are generated from the low-dimensional subspaces, a key intuition of our approach is that we can effectively approximate the solution of (2) using only a small number of selectively sampled data points instead of the whole data matrix $\mathbf{X}$. Specifically, we solve the following problem:

$$\operatorname*{minimize}_{(C_{ji})_{j \in [N]} \in \mathbb{R}^N} \frac{1}{2} \left\| \mathbf{x}_i - \sum_{j \in [N]} C_{ji} \mathbf{x}_j \right\|_2^2 + \lambda \sum_{j \in [N]} |C_{ji}|, \text{ subject to } C_{ji} = 0, \forall j \in \{i\} \cup ([N] \setminus S),$$

$$\tag{3}$$

where $S \subset [N]$ denotes indices of selected subsamples. This problem can be solved by standard solvers of $\ell_1$ minimization problem, such as GLMNET [31] and coordinate descent methods [32] by time and space complexity independent of $N$. The key challenge of the approach is to obtain the set of subsamples $S$ such that all subspaces are sufficiently covered and the obtained solution is close to the global solution of (1).

To solve this challenge, we propose an incremental algorithm for obtaining $S$ based on the stochastic approximation of a subgradient. Let us assume that $(C_{ji}^S)_{ji} = \mathbf{C}^S$ is formed by the optimal solutions of (3) and we need to find the next data point $i_+ \in [N] \setminus S$ so that $\mathbf{C}^{S \cup \{i_+\}}$ is close to the optimal solution of (1). Our strategy is to choose next $i_+$ in terms of the most violating subgradient. We explain below how we define the violation and compute it efficiently.

First, let $G_{ji}$ be the subdifferential of the objective function in (2) with respect to $C_{ji}^S$. Then, a necessary condition for the objective function of (2) not to decrease by newly adding $i' \in [N] \setminus S$ to $S$ can be written as $G_{i'i} \ni 0$, for all $i \in [N] \setminus \{i'\}$. Here, the subdifferential $G_{i'i}$ is given by the following equation:

$$
G_{i'i} = \begin{cases} \left\langle \mathbf{x}_{i'}, \sum_{j \in [N]} C_{ji}^S \mathbf{x}_j - \mathbf{x}_i \right\rangle + [-\lambda, \lambda] & C_{i'i}^S = 0, \\ \left\langle \mathbf{x}_{i'}, \sum_{j \in [N]} C_{ji}^S \mathbf{x}_j - \mathbf{x}_i \right\rangle + \mathrm{sign}\left(C_{i'i}^S\right) \lambda & \text{otherwise.} \end{cases}
$$

Then, the necessary condition can be written as follows:

$$
G_{i'i} \ni 0 \Leftrightarrow med \left\{ 0, \left\langle \mathbf{x}_{i'}, \sum_{j \in [N]} C_{ji}^S \mathbf{x}_j - \mathbf{x}_i \right\rangle \pm \lambda \right\} = 0, \tag{4}
$$

where $med$ denotes the median of three values. To assure that adding $i'$-th data point to $S$ always improves the objective function value of (3), the left hand side of (4) has to be non-zero for at least some $i \in [N] \setminus \{i'\}$. Therefore, we measure the violation of the subgradient for each $i' \in [N] \setminus S$ by $\sum_{i \in [N] \setminus \{i'\}} g_{i'i}^2$, where

$$
g_{i'i} = med \left\{ 0, \left\langle \mathbf{x}_{i'}, \sum_{j \in [N]} C_{ji}^S \mathbf{x}_j - \mathbf{x}_i \right\rangle \pm \lambda \right\}. \tag{5}
$$

However, computing (5) for all $(i', i) \in ([N] \setminus S) \times [N]$ requires $O(N^2)$ time. To reduce time complexity, we perform stochastic approximation of the amount $\sum_{i \in [N] \setminus \{i'\}} g_{i'i}^2$. Specifically, we approximate the violation of the subgradient for each $i' \in [N] \setminus S$ using a random subsample $I \subset [N]$ as

$$
\sum_{i \in [N] \setminus \{i'\}} g_{i'i}^2 \approx \frac{N-1}{|I \setminus \{i'\}|} \sum_{i \in I \setminus \{i'\}} g_{i'i}^2,
$$

where $|\cdot|$ denotes cardinality function. Finally, we select $i_+$ as the maximizer of the right hand side among $i' \in [N] \setminus S$, which can be computed in $O(|I|N)$, where $|I| \ll N$ and can be considered as a constant. In all experiments and analyses, we use only one random subsample, i.e., $|I| = 1$. Since this is not time critical step, using more subsamples benefits the algorithm. Pseudocode of representation learning step is summarized in Algorithm 1.

## 3.2 Spectral clustering

Given sparse affinity matrix $\mathbf{W}$, S$^5$C algorithm effciently solves spectral clustering step by performing eigenvalue decomposition using orthogonal iteration. Power method is a well known approach for approximating dominant eigenvector by iterative matrix-vector multiplication. Orthogonal iteration computes eigenvectors in a block by iteratively performing matrix-matrix multiplication and orthogonalization of the block using QR factorization. In the general setting, orthogonal iteration suffers from $O(N^2)$ computational complexity. On the other hand, orthogonal iteration in our setting enjoys $O(N)$ scalability. This is achieved by the guarantee that $\mathbf{W}$ contains non-zero elements linear in the number of data points $N$.

Spectral clustering algorithm requires computation of $L$ eigenvectors associated with the $L$ smallest eigenvalues of symmetric normalized Laplacian matrix $\mathbf{L}_S$. They can be found as the largest eigenvectors of the positive semidefinite matrix $2\mathbf{I}_N - \mathbf{L}_S$, where $2$ comes from the upper-bound

---

**Algorithm 1** Representation learning step in S$^5$C

---

**Require:** Dataset $(\mathbf{x}_1, \ldots, \mathbf{x}_N) \in \mathbb{R}^{M \times N}$, hyperparameter $\lambda$, number of iterations $T$, batch size $B$
1: $\quad S \leftarrow \varnothing$
2: **for** $t \in [T]$ **do**
3: $\qquad$ Randomly sample $I \subset [N]$ such that $|I| = B$
4: $\qquad$ Obtain $(C_{ji})_{j \in [N]}$ by solving (3) for $i \in I$
5: $\qquad g_{i'i} \leftarrow med \left\{ 0, \left\langle \mathbf{x}_{i'}, \sum_{j \in S \setminus \{i\}} C_{ji} \mathbf{x}_j - \mathbf{x}_i \right\rangle \pm \lambda \right\}$ for $(i', i) \in ([N] \setminus S) \times I$
6: $\qquad i_+ \leftarrow \text{argmax}_{i' \in [N] \setminus S} \frac{N-1}{|I \setminus \{i'\}|} \sum_{i \in I \setminus \{i'\}} g_{i'i}^2$
7: $\qquad$ **if** $\sum_{i \in I \setminus \{i_+\}} g_{i_+i}^2 \neq 0$ **then**
8: $\qquad\qquad S \leftarrow S \cup \{i_+\}$
9: Obtain $\mathbf{C}$ by solving (3) for all $i \in [N]$
10: $\mathbf{W} \leftarrow |\mathbf{C}| + |\mathbf{C}|^\top$

---

of the eigenvalues of $\mathbf{L}_S$ [33]. We then apply orthogonal iteration to matrix $\mathbf{L}_M$ to find its $L$ largest eigenvectors. We check the convergence condition of orthogonal iteration by evaluating the scaled norm difference between previous and current solutions. Spectral clustering step of S$^5$C is summarized in Algorithm 2 in Appendix A.

## 4 Theoretical guarantees

In this section, we analyze S$^5$C algorithm from the theoretical aspect. We assume $\dim(\mathcal{S}_\ell) = d$ for all $\ell \in [L]$ solely for the simplicity of notation. As established in the literature [14], we provide guarantees on Subspace Detection Property (SDP), which is formally defined as follows.

**Definition 1** (Subspace Detection Property). *An algorithm is said to exhibit subspace detection property if and only if it produces affinity matrix $\mathbf{C} \in \mathbb{R}^{N \times N}$ such that the following conditions hold:*

  1. *For all $i \in [N]$, $i$-th column of $\mathbf{C}$ is not $\mathbf{0}$.*
  2. *For all $i \in [N]$, $i$-th column of $\mathbf{C}$ has non-zero elements only in those rows that correspond to data points that belong to the same subspace as $i$-th data point.*

SDP is known to be guaranteed if SSC is solved with all data points [27, 15], i.e., $|S| = N$ in our notation. In this work, we show that SDP is guaranteed even when $|S| = \tilde{O}(dL + L^2)$, i.e., independent of number of data points $N$. We analyzed S$^5$C algorithm under deterministic data model and random data model. We provide all proofs in Appendices B and C. Our theoretical results can be easily adapted for the case when data points are randomly sampled (Appendix D).

### 4.1 Deterministic data model

In deterministic data model [15], we assume there is no noise but subspaces can intersect in an arbitrary manner. To quantify subspace structure, we introduce two measures: persistent inradius and coherence. Persistent inradius of data points is a measure originally introduced in our work as a useful extension of inradius of data points [14] and quantifies how much data points are uniformly distributed in each subspace. Figure 1 illustrates the idea of persistent inradius in the low-dimenional space. Coherence [23] is a measure which quantifies closeness between two subspaces.

**Definition 2** (Inradius). *The inradius of convex body $P$, denoted by $r(P)$, is defined as the radius of the largest Euclidean ball inscribed in $P$.*

**Definition 3** (Persistent inradius). *The persistent inradius with respect to $P = \{P_i\}_{i \in [m]} \subset \mathbb{R}^d$, denoted by $\check{r}(P)$, is defined as the minimum inradius of symmetric convex bodies represented as $\text{conv}\left(\{\pm P_i\}_{i \in I}\right)$, where $|I| \geq d$.*

**Definition 4** (Coherence). *The coherence $\mu(X, Y)$ between two sets of points of unit norm, $X$ and $Y$, is defined as*

$$\mu(X, Y) = \max_{\mathbf{x} \in X, \mathbf{y} \in Y} \langle \mathbf{x}, \mathbf{y} \rangle .$$

**Theorem 1.** *Assume that data* $\mathbf{X} \in \mathbb{R}^{M \times N}$ *with normalized columns and subspaces* $\{\mathcal{S}_\ell\}_{\ell \in [L]}$ *are given. We define* $\ell(i)$ *so that the subspace corresponding to* $i$*-th data is* $\mathcal{S}_{\ell(i)}$ *and* $S_\ell = \{i \in [N] | \ell(i) = \ell\}$*. Assume that* $|S_\ell| = N/L$ *and* $\dim \mathcal{S}_\ell = d$*, for all* $\ell \in [L]$*.* $\mathbf{X}[S_\ell]$ *denotes the subset which corresponds to data in* $S_\ell$*. We define*

$$\check{r} = \min_\ell \check{r}(\mathbf{X}[S_\ell]), \ \mu = \max_{\ell \neq \ell'} \mu\left(\mathbf{X}[S_\ell], \mathbf{X}[S_{\ell'}]\right).$$

*If it holds that*

$$0 < \mu \leq \lambda < \check{r}, \ T \geq 2\left(1 + \frac{L}{d}\left(\log(2L\delta^{-1})\right)\right) dL,$$

*then,* S$^5$C *of* $T$ *iterations with hyerparameter* $\lambda$ *has subspace detection property with at least probability* $1 - \delta$*.*

This theorem implies that if $\mu < \check{r}$ holds, there exists hyperparameter $\lambda$ that makes S$^5$C able to exactly recover subspaces. The randomness in the model is introduced with the random selection of subsample $I$ (line 3 of Algorithm 1). Theorem also provides approximation guarantees by implying that the number of iterations sufficient for S$^5$C to obtain SDP with high probability is independent of $N$. Note that S$^5$C chooses subsample only if the condition in line 7 of Algorithm 1 is satisfied, meaning that less than $T$ subsamples can be sufficient for the algorithm to obtain SDP. Therefore, number of iterations $T$ is linearly connected to the runtime of the algorithm and has an interpretation as an upper bound on the number of subsamples $|S|$. In the case when subsamples are randomly chosen, we can easily extend the proof and show that $T$ ran-

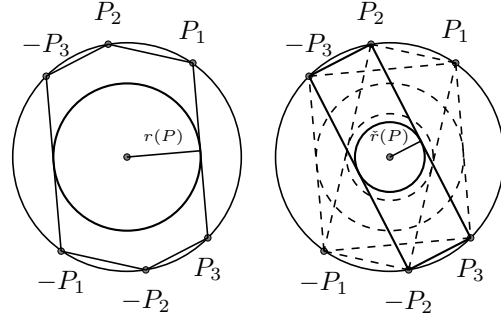

Figure 1: Concept of persistent inradius. Left: inradius $r\left(\mathrm{conv}(P)\right)$ where $P = \{\pm P_1, \pm P_2, \pm P_3\}$. Right: persistent inradius $\check{r}(P)$.

domly chosen subsamples can also enjoy SDP with high probability. However, in this case the number of iterations $T$ corresponds to the number of subsamples. Therefore, theory implies that S$^5$C may need less number of subsamples to satisfy SDP compared to random subsamples. In the case when the number of dimensions varies among the subspaces, it is straightforward to generalize the theorem by setting $d = \max_{\ell \in [L]} d_\ell$.

## 4.2 Random data model

We introduce semi-random model [14, 15] for our analysis of random data model.

**Definition 5** (Semi-random model). *Data* $\mathbf{X}$ *is drawn from semi-random model if and only if, for each* $\ell$*, each element of* $\mathbf{X}[S_\ell]$ *is drawn from uniform distribution on the surface of the unit ball with respect to the subspace* $\mathcal{S}_\ell$*.*

To measure the closeness between two subspaces under the random data model, we introduce affinity.

**Definition 6** (Affinity). *Affinity between two* $d$*-dimensional subspaces* $\mathcal{S}$ *and* $\mathcal{S}'$ *in* $\mathbb{R}^M$ *denoted by* $\mathrm{aff}(\mathcal{S}, \mathcal{S}')$ *is defined as follows:*

$$\mathrm{aff}(\mathcal{S}, \mathcal{S}') = \max_{\mathbf{U} \in O(\mathcal{S})} \max_{\mathbf{V} \in O(\mathcal{S}')} \left\|\mathbf{U}^\top \mathbf{V}\right\|_F,$$

*where* $O(\mathcal{S})$ *denotes the set of matrices which induces projection onto* $\mathcal{S}$*, i.e.,* $O(\mathcal{S}) = \left\{\mathbf{V} = (\mathbf{v}_j)_j \in \mathbb{R}^{d \times M} \ \middle| \ \mathbf{v}_j \in \mathcal{S}, \langle \mathbf{v}_i, \mathbf{v}_j \rangle = \delta_{ij}\right\}$*.*

An alternative definition of affinity in terms of principal angles can be found in [14, 15].

**Theorem 2.** *Assume that data* $\mathbf{X} \in \mathbb{R}^{M \times N}$ *is drawn from semi-random model in which subspaces* $\{\mathcal{S}_\ell\}_{\ell \in [L]}$ *are given. We define*

$$\rho = \frac{N}{dL}, \ a = \min_{\ell \neq \ell'} \mathrm{aff}(\mathcal{S}_\ell, \mathcal{S}_{\ell'}).$$

*If it holds that*

$$4 < \log \rho < 4d, \ a \le \lambda < \frac{1}{8}\sqrt{\frac{\log \rho}{d}}, \ T \ge 2\left(1 + \frac{L}{d}\left(\log(2L\delta^{-1})\right)\right)dL, \qquad (6)$$

*then,* $S^5C$ *of $T$ iterations with hyperparameter $\lambda$ has subspace detection property with at least probability* $1 - \delta - L\exp(-d\sqrt{\rho})$.

This theorem implies that if conditions (6) on $\rho$, $\lambda$ and $T$ hold, then $S^5C$ satisfies SDP with high probability. This theorem can also be easily adapted for the case of randomly selected data points.

## 5 Experimental evaluation

**Baselines and evaluation metrics.** We compare clustering performance and scalability to other SSC based methods, including Sparse Subspace Clustering (SSC) [7], Scalable Sparse Subspace Clustering (SSSC) [19, 20], Sparse Subspace Clustering via Orthogonal Matching Pursuit (SSC-OMP) [22] and Elastic Net Subspace Clustering with ORacle Guided Elastic Net (EnSC-ORGEN) [27]. Besides sparse subspace clustering methods, we compare performance to Nyström algorithm [34] and Approximate Kernel $K$-means (AKK) [35]. Our code is available at `https://github.com/smatsus/S5C`. Clustering performance is evaluated in terms of the clustering error (CE) defined as $CE(\hat{\mathbf{r}}, \mathbf{r}) = \min_{\pi \in \Pi_L}\left(1 - \frac{1}{N}\sum_{i \in [N]}\mathbf{1}_{\{\pi(\hat{\mathbf{r}}_i) = \mathbf{r}_i\}}\right)$, where $\Pi_L$ is the set of all permutations on $[L]$.

**Benchmark datasets.** We verify the effectiveness of $S^5C$ on six benchmark datasets including face image dataset Yale B [36, 37], motion segmentation Hopkins 155 [38], object recognition datasets COIL-100 [39] and CIFAR-10 [40], handwritten digits dataset MNIST [41], letter recognition dataset of different fonts Letter-rec [42], and handwritten character recognition dataset Devanagari [43]. The summary of datasets and details of experimental setup are provided in Appendix E.

**Clustering performance.** Clustering error of $S^5C$ algorithm compared to the state-of-the-art methods on six real-world datasets is presented in Table 2. The results show that $S^5C$ is the only algorithm which consistently has good performance, achieving 13% better median performance over the second best SSC-ORGEN. On the COIL-100 dataset which has 100 classes, $S^5C$ achieves score close to the SSC baseline and significantly outperforms all other methods. In all experiments, we use only one random subsample, i.e., $|I| = 1$. In order to examine the sensitivity of $S^5C$ to the random sampling line 3 in Algorithm 1, we rerun the algorithm with different random seeds and report means and standard deviations over 10 runs. The results demonstrate that $S^5C$ is not sensitive to this step and standard deviation varies from 0.4 % to 2.3 % across all datasets.

Table 2: Clustering error (%): Character '/' denotes that either time limit of 24 hours or memory limit of 16 GB was exceeded. Standard deviations of $S^5C$ are given in parentheses.

| Dataset | Nyström | AKK | SSC | SSC-OMP | SSC-ORGEN | SSSC | $S^5C$ |
|---|---|---|---|---|---|---|---|
| Yale B | 76.8 | 85.7 | 33.8 | 35.9 | 37.4 | 59.6 | 39.3 (1.8) |
| Hopkins 155 | 21.8 | 20.6 | 4.1 | 23.0 | 20.5 | 21.1 | 14.6 (0.4) |
| COIL-100 | 54.5 | 53.1 | 42.5 | 57.9 | 89.7 | 67.8 | 45.9 (0.5) |
| Letter-rec | 73.3 | 71.7 | / | 95.2 | 68.6 | 68.4 | 67.7 (1.3) |
| CIFAR-10 | 76.6 | 75.6 | / | / | 82.4 | 82.4 | 75.1 (0.8) |
| MNIST | 45.7 | 44.6 | / | / | 28.7 | 48.7 | 40.4 (2.3) |
| Devanagari | 73.5 | 72.8 | / | / | 58.6 | 84.9 | 67.2 (1.3) |

**Computational time.** We compare computational time to other large-scale methods using randomly sampled subsets on the COIL-100 and MNIST datasets. Figures 2 (a) and (b) show the mean computational time for each cardinality of independents subsets. As expected by theory, computational time of $S^5C$ increases only linearly with the respect to the number of data points. Most of the time of $S^5C$ is taken by solving LASSO, which is extremely easy to parallelize just by partitioning data points across machines. We do not focus on such implementation improvements as our point here is not in reporting faster time, but in showing the linear scalability and consequently the ability to handle large-scale data.

**Benefits of selective sampling.** The main motivation behind the selective sampling in the representation learning step is to better capture structure of the entire dataset than simple random sampling. To

evaluate this hypothesis, we design an experiment which compares the performance of subsamples selected based on the stochastic approximation of the subgradient to random subsamples. For this purpose, we consider a method in which the selective sampling in the representation learning step is replaced with random sampling. We call this method S$^5$C-rand. Figure 2 (c) shows the objective function value with respect to the number of subsamples achieved by S$^5$C and S$^5$C-rand methods on the Yale B dataset. It can be seen that for each subsample S$^5$C achieves lower value of the objective function. Furthermore, by using $\sim 75\%$ of subsamples S$^5$C achieves the same objective function value as S$^5$C-rand.

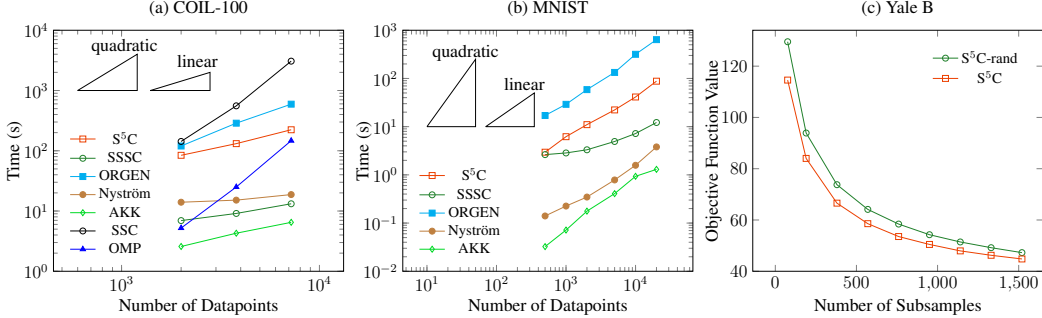

Figure 2: (a) and (b) Relation between training time and number of datapoints on the COIL-100 and MNIST datasets. (c) Objective function value of selective and random sampling based S$^5$C on the Yale B dataset.

**Benefits of orthogonal iteration.** We further compare performance and time efficiency of the spectral clustering step in S$^5$C with a classic eigenvalue decomposition algorithm for the normalized cut (NCut) referred to as NCutE in [26], and Power Iteration Clustering (PIC) [26]. In PIC method authors use power method to find the dominant eigenvector and then apply $K$-means clustering to one-dimensional vector. We call our method Orthogonal Iteration Clustering (OIC). We design the experiment so that each of the algorithms receives the same affinity matrix $\mathbf{W}$ at the input obtained by S$^5$C representation learning step. In this way we compare clustering performance of only spectral decomposition. Since high computational complexity of NCut limits the application to large-datasets, we compare performance on Yale B and COIL-100 datasets. To avoid that the computational time is dominated by $K$-means clustering, we report time obtained with only one execution of $K$-means, while in practice it is often executed several times with different initializations. The results are shown in Table 3. The experiments demonstrate that OIC performs comparably to NCut and does not degrade clustering performance. Although PIC has lower computational time than OIC, it fails to provide satisfying clustering accuracy.

Table 3: Clustering error (CE) and computational time of spectral clustering step on the Yale B and COIL-100 datasets.

| Dataset | Measure | NCut | PIC | OIC |
|---------|---------|------|-----|-----|
| Yale B | CE (%) | 42.7 | 83.9 | 42.8 |
|  | Time (s) | 38.7 | 12.0 | 16.2 |
| COIL-100 | CE (%) | 47.0 | 77.4 | 45.4 |
|  | Time (s) | 290.0 | 16.4 | 27.4 |

# 6   Conclusion

Building on the existing work on sparse subspace clustering (SSC), this paper introduced the efficient SSC algorithm, called S$^5$C, able to linearly scale to the number of data points in both representation learning and spectral clustering steps. We derived theoretical conditions under which subspace detection property of S$^5$C is preserved. Besides computational efficiency, experimental results showed that S$^5$C achieves performance improvement over existing large-scale sparse subspace clustering algorithms. Our algorithm is not restricted to SSC but can be easily extended to elastic net subspace clustering. We believe our approach will expand the applicability of sparse subspace clustering algorithm to large-scale datasets.

**Acknowledgments**

This work was supported by KAKENHI 19K20336.

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
