[Supplementary Material]

# Appendices to
# Selective Sampling-based Scalable Sparse Subspace Clustering

**Shin Matsushima**
University of Tokyo
smatsus@graco.c.u-tokyo.ac.jp

**Maria Brbić**
Stanford University
mbrbic@cs.stanford.edu

**Outline**. Appendix A provides pseudocodes of representation learning and spectral clustering steps of S$^5$C and analysis of time and memory complexity. In Appendix B, we present results on deterministic data model that lead to the proof of Theorem 1. In Appendix C, we present results on random data model that lead to the proof of Theorem 2. In Appendix D, we show how to adjust both theorems to the case when data points are randomly selected. Details of experimental setup are given in Appendix E. We assume $\dim(\mathcal{S}_\ell) = d$ for all $\ell \in L$ only for the simplicity of notation.

## Appendix A: S$^5$C algorithm

### Pseudocodes

We restate the optimization problem of S$^5$C:

$$\operatorname*{minimize}_{(C_{ji})_{j \in [N]} \in \mathbb{R}^N} \frac{1}{2} \left\| \mathbf{x}_i - \sum_{j \in [N]} C_{ji} \mathbf{x}_j \right\|_2^2 + \lambda \sum_{j \in [N]} |C_{ji}|, \text{ subject to } C_{ji} = 0, \forall j \in \{i\} \cup ([N] \setminus S), \tag{1}$$

where $S \subset [N]$ denotes indices of selected subsamples. For completeness, pseudocode of representation learning step of S$^5$C is given in Algorithm 1. Pseudocode of spectral clustering step is given in Algorithm 2.

### Time and memory complexity

First, we consider computational cost in the selective sampling part of the Algorithm 1 (lines 2-8) which solves $B$ LASSO problems for each $t \in [T]$, where $B$ is cardinality of $I$, and $T$ is the number of iterations. Each LASSO problem consists of $O(|S|)$ parameters and the design matrix of $O(|S|M)$ elements. By utilizing standard solvers of LASSO based on coordinate descent methods such as GLMNET [1] with a fixed tolerance, the solution can be obtained in $O(|S|M)$ computational time. Since we solve $BT$ LASSO problems in total and $|S| = O(T)$, resulting computational time is $O(T^2BM)$. Memory complexity amounts to the memory required for each LASSO problem and takes only $O(TM)$ space. Computing $g_{i'i}$ takes $O(NBT)$ time complexity and $O(NB)$ space complexity. Finally, in line 9 of Algorithm 1, we solve $N$ LASSO problems that consist of $O(T)$ parameters and the design matrix with $O(TM)$ elements, which requires dominant computational time and space. However, this step is very easy to parallelize. In total, in the representation learning step S$^5$C method has $O((TB + N)TM)$ time and $O(TM + NB)$ memory complexity in total.

In the spectral clustering step, the key observation to reduce computation is that $\mathbf{C}$ and hence, $\mathbf{W} = |\mathbf{C}| + |\mathbf{C}|^\top$ have only $O(TN)$ number of non-zero elements. As a result, a matrix-vector product can be performed in $O(TN)$ time and space complexity. Therefore, we only require $O((L + T)N)$ for computing $L$ eigenvectors. We note that step 2 of Algorithm 2 which computes

---
**Algorithm 1** Representation learning step in S⁵C
---
**Require:** Dataset $(\mathbf{x}_1, \ldots, \mathbf{x}_N) \in \mathbb{R}^{M \times N}$, hyperparameter $\lambda$, number of iterations $T$, batch size $B$
1: $S \leftarrow \varnothing$
2: **for** $t \in [T]$ **do**
3:      Randomly sample $I \subset [N]$ such that $|I| = B$
4:      Obtain $(C_{ji})_{j \in [N]}$ by solving (1) for $i \in I$
5:      $g_{i'i} \leftarrow med \left\{ 0, \left\langle \mathbf{x}_{i'}, \sum_{j \in S \setminus \{i\}} C_{ji} \mathbf{x}_j - \mathbf{x}_i \right\rangle \pm \lambda \right\}$ for $(i', i) \in ([N] \setminus S) \times I$
6:      $i_+ \leftarrow \operatorname{argmax}_{i' \in [N] \setminus S} \frac{N-1}{|I \setminus \{i'\}|} \sum_{i \in I \setminus \{i'\}} g_{i'i}^2$
7:      **if** $\sum_{i \in I \setminus \{i_+\}} g_{i_+ i}^2 \neq 0$ **then**
8:          $S \leftarrow S \cup \{i_+\}$
9: Obtain $\mathbf{C}$ by solving (1) for all $i \in [N]$
10: $\mathbf{W} \leftarrow |\mathbf{C}| + |\mathbf{C}|^\top$
---

---
**Algorithm 2** Spectral clustering step in S⁵C
---
**Require:** Affinity matrix $\mathbf{W} \in \mathbb{R}^{N \times N}$, number of clusters $L \in \mathbb{N}$, error tolerance $\epsilon \in \mathbb{R}$,
1: $\mathbf{D} \leftarrow \operatorname{diag}(\mathbf{W}\mathbf{1})$
2: $\mathbf{L}_\mathrm{S} \leftarrow \mathbf{I}_N - \mathbf{D}^{-\frac{1}{2}} \mathbf{W} \mathbf{D}^{-\frac{1}{2}}$
3: $\mathbf{L}_M \leftarrow 2\mathbf{I}_N - \mathbf{L}_S$
4: Initialize $\mathbf{V} \in \mathbb{R}^{N \times L}$ such that $\mathbf{V}^\top \mathbf{V} = \mathbf{I}_L$
5: **while** $\|\mathbf{V} - \mathbf{V}_{\mathrm{prev}}\|_F / \sqrt{LN} \geq \epsilon$ **do**
6:      $\mathbf{V}_{\mathrm{prev}} \leftarrow \mathbf{V}$
7:      $\mathbf{V} \leftarrow \mathbf{L}_M \mathbf{V}$
8:      Orthogonalize $\mathbf{V}$ by QR factorization
9: Normalize each row of $\mathbf{V}$ to unit length
10: Apply $K$-means clustering on $\mathbf{V}$
---

Laplacian matrix is not dominant in terms of both time and space complexity, as computing $\mathbf{D}$ requires only $O(TN)$ time and the resulting matrix can be expressed by $N$-dimensional vector. In order to produce eigenvectors in lines 5-8 matrix multiplication and QR decomposition have $O(TNL)$ and $O(NL^2)$ time in each iteration, respectively. Finally, with a fixed tolerance $\epsilon$, required time is $O\left(\log \epsilon^{-1}\right)$ iterations. Therefore, spectral clustering step has $O\left((T+L)LN \log \epsilon^{-1}\right)$ time and $O(TN)$ space complexity.

## Appendix B: Deterministic data model

For the completeness, we first restate Theorem 1.

**Theorem 1.** *Assume that data* $\mathbf{X} \in \mathbb{R}^{M \times N}$ *with normalized columns and subspaces* $\{\mathcal{S}_\ell\}_{\ell \in [L]}$ *are given. We define* $\ell(i)$ *so that the subspace corresponding to $i$-th data is* $\mathcal{S}_{\ell(i)}$ *and* $S_\ell = \{i \in [N] | \ell(i) = \ell\}$. *Assume that* $|S_\ell| = N/L$ *and* $\dim \mathcal{S}_\ell = d$, *for all* $\ell \in [L]$. $\mathbf{X}[S_\ell]$ *denotes the subset which corresponds to data in* $S_\ell$. *We define*

$$\check{r} = \min_\ell \check{r}(\mathbf{X}[S_\ell]), \ \mu = \max_{\ell \neq \ell'} \mu\left(\mathbf{X}[S_\ell], \mathbf{X}[S_{\ell'}]\right). \tag{2}$$

*If it holds that*

$$0 < \mu \leq \lambda < \check{r}, \ T \geq 2\left(1 + \frac{L}{d}\left(\log(2L\delta^{-1})\right)\right) dL, \tag{3}$$

*then,* S⁵C *of $T$ iterations with hyperparameter $\lambda$ has subspace detection property with at least probability* $1 - \delta$.

The proof of the theorem follows from three lemmas which we state and prove below. All lemmas use the same assumptions and definitions from the statement of Theorem 1.

**Lemma 1.** *Define for any $S \subset [N]$ and $i \in [N]$,*

$$\mathbf{c}^S = (c_j^S)_{j\in[N]} = \underset{(c_j)_{j\in[N]}\in\mathbb{R}^N}{\mathrm{argmin}} \frac{1}{2}\left\|\mathbf{x}_i - \sum_{j\in S\setminus\{i\}} c_j\mathbf{x}_j\right\|_2^2 + \lambda \sum_{j\in S\setminus\{i\}}|c_j|, \tag{4}$$

*where $\mathbf{x}_i$ is $i$-th column of $\mathbf{X}$. If it holds that*

$$\mu \leq \lambda, \tag{5}$$

*then, $\mathbf{c}^{S\cap S_{\ell(i)}}$ induces $\mathbf{c}^S$, i.e.,*

$$c_j^S = \begin{cases} c_j^{S\cap S_{\ell(i)}} & j \in S_{\ell(i)}, \\ 0 & j \notin S_{\ell(i)}. \end{cases} \tag{6}$$

*Proof.* For any $S \subset [N]$ and $i \in [N]$, define $\mathbf{c} = (c_j)_j$ as follows:

$$c_j = \begin{cases} c_j^{S\cap S_{\ell(i)}} & j \in S_{\ell(i)}, \\ 0 & j \notin S_{\ell(i)}. \end{cases} \tag{7}$$

Then, for the lemma to be true, $\mathbf{c}$ has to satisfy the following optimality condition

$$\left|\left\langle \mathbf{x}_{i'}, \sum_{j\in S\setminus\{i\}} c_j\mathbf{x}_j - \mathbf{x}_i \right\rangle\right| \leq \lambda, \tag{8}$$

for all $i' \in S$ such that $\ell(i') \neq \ell(i)$. This follows from

$$\left|\left\langle \mathbf{x}_{i'}, \sum_{j\in S\setminus\{i\}} c_j\mathbf{x}_j - \mathbf{x}_i \right\rangle\right| = \left|\left\langle \mathbf{x}_{i'}, \sum_{j\in\left(S\cap S_{\ell(i)}\right)\setminus\{i\}} c_j^{S\cap S_{\ell(i)}}\mathbf{x}_j - \mathbf{x}_i \right\rangle\right| \tag{9}$$

$$\leq \mu\left\|\sum_{j\in\left(S\cap S_{\ell(i)}\right)\setminus\{i\}} c_j^{S\cap S_{\ell(i)}}\mathbf{x}_j - \mathbf{x}_i\right\|_2 \tag{10}$$

$$\leq \mu \leq \lambda. \tag{11}$$

The second last inequality holds from the following inequality.

$$\frac{1}{2}\left\|\mathbf{x}_i - \sum_{j\in\left(S\cap S_{\ell(i)}\right)\setminus\{i\}} c_j^{S\cap S_{\ell(i)}}\mathbf{x}_j\right\|_2^2 \tag{12}$$

$$\leq \frac{1}{2}\left\|\mathbf{x}_i - \sum_{j\in\left(S\cap S_{\ell(i)}\right)\setminus\{i\}} c_j^{S\cap S_{\ell(i)}}\mathbf{x}_j\right\|_2^2 + \lambda \sum_{j\in\left(S\cap S_{\ell(i)}\right)\setminus\{i\}}\left|c_j^{S\cap S_{\ell(i)}}\right| \tag{13}$$

$$\leq \frac{1}{2}\|\mathbf{x}_i - \mathbf{0}\|_2^2 + \lambda \sum_{j\in S\setminus\{i\}}|0| = \frac{1}{2}. \tag{14}$$

Therefore, $\mathbf{c}$ defined in (7) is the optimal solution $\mathbf{c}^S$. □

**Lemma 2.** *Let singleton $\{i_t\}$ be the set of random samples $I$ and $S_t$ be the set of selected subsamples $S$ at $t$-th iteration of Algorithm 1 of $\mathrm{S}^5\mathrm{C}$. For $t \in [T]$, $t$-th iteration of $\mathrm{S}^5\mathrm{C}$ is said to be* expanding *if and only if the following is satisfied:*

$$\sum_\ell \dim(\mathcal{S}_\ell \cap \mathrm{span}(S_{t+1})) = 1 + \sum_\ell \dim(\mathcal{S}_\ell \cap \mathrm{span}(S_t)). \tag{15}$$

*If $i_t$ satisfies that $\left|S_{\ell(i_t)} \cap S_t\right| < d$, then $t$-th iteration of $\mathrm{S}^5\mathrm{C}$ is expanding.*

*Proof.* At $t$-th iterations of Algorithm 1, the subsample to be added is expressed as

$$\underset{i'}{\operatorname{argmax}} \left| med \left\{ \left\langle \mathbf{x}_{i'}, \mathbf{r}_{i_t}^{S_t} \right\rangle \pm \lambda, 0 \right\} \right|,$$

where

$$\mathbf{r}_{i_t}^{S_t} = \sum_{j \in S_t \setminus \{i_t\}} c_{ji_t}^{S_t} \mathbf{x}_j - \mathbf{x}_{i_t}, \tag{16}$$

$$\mathbf{c}_{i_t}^{S_t} = (c_{ji_t}^{S_t})_{j \in [N]} = \underset{(c_j)_j \in \mathbb{R}^N}{\operatorname{argmin}} \frac{1}{2} \left\| \mathbf{x}_{i_t} - \sum_{j \in S_t \setminus \{i_t\}} c_j \mathbf{x}_j \right\|_2^2 + \lambda \sum_{j \in S_t \setminus \{i_t\}} |c_j|. \tag{17}$$

Therefore, it is sufficient to show

$$\max_{i'} \left| med \left\{ \left\langle \mathbf{x}_{i'}, \mathbf{r}_{i_t}^{S_t} \right\rangle \pm \lambda, 0 \right\} \right| > 0, \tag{18}$$

and

$$\underset{i'}{\operatorname{argmax}} \left| med \left\{ \left\langle \mathbf{x}_{i'}, \mathbf{r}_{i_t}^{S_t} \right\rangle \pm \lambda, 0 \right\} \right| \in S_{\ell(i_t)}. \tag{19}$$

By Lemma 1, $\mathbf{c}_{i'}^{S_t \cap S_{\ell(i)}}$ induces $\mathbf{c}_{i'}^{S_t}$ for any $i'$, which means that $c_{i'i_t}^{S_t} = 0$ and that $\left| med \left\{ \left\langle \mathbf{x}_{i'}, \mathbf{r}_{i_t}^{S_t} \right\rangle \pm \lambda, 0 \right\} \right| = 0$ for $i' \in S_t \setminus S_{\ell(i_t)}$. This means that any $i'$ such that $\ell(i') \neq \ell(i_t)$ is not chosen as maximizer in (19).

Now we show there exists $j$ such that

$$\left| \left\langle \mathbf{x}_j, \mathbf{r}_{i_t}^{S_t} \right\rangle \right| > \lambda, \tag{20}$$

and that $\ell(j) = \ell(i_t)$. The assumption $\check{r} > 0$ means that any $d$ points in $S_\ell$ are linearly independent. Therefore, $\left| S_{\ell(i_t)} \cap S_t \right| < d$ immediately implies that

$$\dim(\mathcal{S}_\ell \cap \operatorname{span}(S_t \cup \{j\})) = \dim(\mathcal{S}_\ell \cap \operatorname{span}(S_t)) + 1, \tag{21}$$

for any $j \in S_{\ell(i_t)} \setminus S_t$. For any set $J \subset S_{\ell(i_t)}$ such that $\left| (S_{\ell(i_t)} \cap S_t) \cup J \right| = d$ and $(b_j)_{j \in (S_{\ell(i)} \cap S_t) \cup J} \in \{+1, -1\}^d$, $\{b_j \mathbf{x}_j\}_{j \in (S_{\ell(i_t)} \cap S_t) \cup J}$ forms a facet in $\mathcal{S}_{\ell(i_t)}$. By definition of $\check{r}$, we see

$$\left| \left\langle \mathbf{x}_{j'}, \sum_{j \in (S_{\ell(i_t)} \cap S_t) \cup J} \alpha_j \mathbf{x}_j \right\rangle \right| \geq \check{r}, \tag{22}$$

for any $\alpha_j$ such that $\sum_{j \in (S_{\ell(i_t)} \cap S_t) \cup J} \alpha_j = 1, \alpha_j \geq 0$. Therefore,

$$\left| \left\langle \mathbf{x}_{j'}, \mathbf{r}_{i_t}^{S^t} \right\rangle \right| = \left| \left\langle \mathbf{x}_{j'}, \sum_{j \in S_t \setminus \{i_t\}} c_{ji_t}^{S_t} \mathbf{x}_j - \mathbf{x}_{i_t} \right\rangle \right| \tag{23}$$

$$= \left| \left\langle \mathbf{x}_{j'}, \sum_{j \in (S_t \cap S_{\ell(i_t)}) \setminus \{i_t\}} c_{ji_t}^{S_t} \mathbf{x}_j - \mathbf{x}_{i_t} \right\rangle \right| \tag{24}$$

$$\geq \left( \sum_{j \in (S_t \cap S_{\ell(i_t)}) \setminus \{i_t\}} \left| c_{ji_t}^{S_t} \right| + 1 \right) \check{r} \geq \check{r} > \lambda. \tag{25}$$

Therefore, (20) holds and it implies (18) and (19). $\qquad \square$

**Lemma 3.** *Assume that $|S_\ell| = N/L$ and $\dim \mathcal{S}_\ell = d$, for all $\ell \in [L]$ and that $T$ is large enough to satisfy $T \geq 2 \left( 1 + \frac{L}{d} \left( \log(2L\delta^{-1}) \right) \right) dL$. Let $I_T = \{i_t\}_{t \in [T]}$, where singleton $\{i_t\}$ is the set of random samples $I$ at $t$-th iteration of Algorithm 1 of $S^5C$. Then, the probability of event $\{\min_\ell |S_\ell \cap I_T| \geq d\}$ is at least $1 - \delta$.*

*Proof.*

$$\mathbb{P}\left\{\min_{\ell\in[L]}|S_\ell\cap I_T|\geq d\right\}=1-\mathbb{P}\left\{\min_{\ell\in[L]}|S_\ell\cap I_T|<d\right\} \tag{26}$$

$$=1-\mathbb{P}\bigcup_{\ell\in[L]}\{|S_\ell\cap I_T|<d\} \tag{27}$$

$$\geq 1-L\mathbb{P}\{|S_\ell\cap I_T|<d\}, \tag{28}$$

for any $\ell\in[L]$. For a fixed $\ell\in[L]$, we define random variables $\{\xi_t\}_{t\in[T]}$ as

$$\xi_t=\begin{cases}1 & i_t\in S_\ell \\ 0 & \text{otherwise.}\end{cases} \tag{29}$$

Then $\sum_{t=1}^T\xi_t=|S_\ell\cap I_T|$. $\xi_t$s are independent and has expectation of $L^{-1}$ for all $t$. By Hoeffding's inequality, it follows

$$\mathbb{P}\left\{\left|\frac{\sum_t\xi_t}{T}-L^{-1}\right|\geq\tau\right\}\leq 2\exp\left(-\frac{\tau^2 T}{2}\right). \tag{30}$$

Since $\sum_t\xi_t=|S_\ell\cap I_T|<d$ implies $L^{-1}-\frac{\sum_t\xi_t}{T}\geq\tau$ in which $\tau=L^{-1}-\frac{d}{T}$, we see

$$\mathbb{P}\{|S_\ell\cap I_T|<d\}\leq 2\exp\left(-\frac{\left(L^{-1}-\frac{d}{T}\right)^2 T}{2}\right). \tag{31}$$

Let $\alpha$ be a positive number that satisfies $T=(1+\alpha)dL$. Then, $\left(L^{-1}-\frac{d}{T}\right)^2 T=(1+\alpha)\left(1-\frac{1}{1+\alpha}\right)^2 L^{-1}d\geq(\alpha-1)L^{-1}d$. From the assumption on the size of $T$, we see $\alpha-1\geq 2Ld^{-1}\log(2L\delta^{-1})$. Therefore, $\mathbb{P}\{|S_\ell\cap I_T|<d\}\leq(2L)^{-1}\delta$. Substituting this to (28), we complete the proof. $\qquad\square$

Now we prove Theorem 1.

**Proof of Theorem 1**

*Proof.* From Lemma 2, $t$-th iteration of S⁵C is expanding as long as $S_{\ell(i_t)}<d$. From Lemma 3, iterations are expanding at least $d$ times for each $\ell\in[L]$ with probability at least $1-\delta$. Therefore, $|S_\ell\cap S_T|\geq d$ for all $\ell\in[L]$. Finally, we check SDP of the final solution $i$-th column of which is $\mathbf{c}_i^{S_T}$. For any $i\in[N]$, we see $\mathbf{c}_i^{S_T\cap S_{\ell(i)}}$ induces $\mathbf{c}_i^{S_T}$ as stated in Lemma 1. Therefore, the second condition of SDP holds true. The first condition, $\mathbf{c}_i^{S_T}\neq\mathbf{0}$ for any $i\in[N]$, holds true from the fact that the optimality condition for $\mathbf{c}_i=\mathbf{0}$ does not hold true. It can be seen as follows:

$$\left\|\frac{\partial}{\partial\mathbf{c}_i}\frac{1}{2}\|\mathbf{x}_i-\mathbf{X}_T\mathbf{c}_i\|^2\bigg|_{\mathbf{c}_i=\mathbf{0}}\right\|_\infty=\|\mathbf{X}_T^\top\mathbf{x}_i\|_\infty\geq\max_{j\in S_T\cap S_{\ell(i)}}|\langle\mathbf{x}_j,\mathbf{x}_i\rangle|\geq\check{r}>\lambda. \tag{32}$$

Here $\mathbf{X}_T$ denotes a matrix in which each column is $\mathbf{x}_i$ for $i\in S_T$. Therefore, the output of S⁵C algorithm satisfies SDP with probability at least $1-\delta$. $\qquad\square$

## Appendix C: Random data model

We restate Theorem 2 for completeness.

**Theorem 2.** *Assume that data $\mathbf{X}\in\mathbb{R}^{M\times N}$ is drawn from semi-random model in which subspaces $\{\mathcal{S}_\ell\}_{\ell\in[L]}$ are given. We define*

$$\rho=\frac{N}{dL},\ a=\min_{\ell\neq\ell'}\mathrm{aff}(\mathcal{S}_\ell,\mathcal{S}_{\ell'}). \tag{33}$$

*If it holds that*

$$4 < \log \rho < 4d, \ a \leq \lambda < \frac{1}{8}\sqrt{\frac{\log \rho}{d}}, \ T \geq 2\left(1 + \frac{L}{d}\left(\log(2L\delta^{-1})\right)\right)dL, \tag{34}$$

*then,* $S^5C$ *of* $T$ *iterations with hyperparameter* $\lambda$ *has subspace detection property with at least probability* $1 - \delta - L\exp(-d\sqrt{\rho})$.

The proof of the theorem follows from Theorem 1 and Lemma 4. Lemma 4 uses the same assumptions and definitions from the statement of Theorem 2. The basic structure of this lemma is based on Lemma 3.1 in [2], but also contains the modification introduced in section 7.2.1 in [3].

**Lemma 4.** *Assume that* $\{P_i\}_{i\in[m]}$ *are random vector on* $\mathbb{S}^{d-1}$ *and* $K = \mathrm{conv}(\pm P_1, \ldots, P_m)$. *If* $4 < \log \rho < 4d$, *where* $\rho = \frac{m}{d}$, *then, it holds that*

$$\check{r}(K) > \frac{1}{8}\sqrt{\frac{\log \rho}{d}}, \tag{35}$$

*with probability greater than* $1 - e^{-d\sqrt{\rho}}$.

*Proof.* First observe that, with probability 1, the facets of $K$ are simplices. Let $\{Q_i\}_{i\in[d]} \subset \{\pm P_i\}_{i\in[m]}$ and denote a unit vector orthogonal to $\mathrm{conv}(\{Q_i\}_{i\in[d]})$ by $\theta(\{Q_i\}_{i\in[d]})$.

If $\check{r}(K) > \alpha$ for $\alpha \in (0,1)$, it implies that there exists $\{Q_i\}_{i\in[d]}$ such that $K \subset \left\{x \in \mathbb{S}^{d-1} \middle| \left|\left\langle \theta(\{Q_i\}_{i\in[d]}), x\right\rangle\right| \leq \alpha\right\}$. It further implies that $\left|\left\langle \theta(\{Q_i\}_{i\in[d]}), P_{i'}\right\rangle\right| \leq \alpha$ for all $i' \in [m]$. Therefore,

$$\mathbb{P}\{\check{r}(K) > \alpha\} \leq \mathbb{P}\bigcup_{\{Q_i\}_{i\in[d]}}\bigcap_{i'\in[m]}\left\{\left|\left\langle\theta(\{Q_i\}_{i\in[d]}), P_{i'}\right\rangle\right| \leq \alpha\right\}. \tag{36}$$

For fixed $\{Q_i\}_{i\in[d]}$, let $I = \left\{i' \in [m] \middle| P_{i'} \in \{Q_i\}_{i\in[d]} \text{ or } -P_{i'} \in \{Q_i\}_{i\in[d]}\right\}$. Then,

$$\mathbb{P}\bigcap_{i'\in[m]}\left\{\left|\left\langle\theta(\{Q_i\}_{i\in[d]}), P_{i'}\right\rangle\right| \leq \alpha\right\} \leq \mathbb{P}\bigcap_{i'\in[m]\setminus I}\left\{\left|\left\langle\theta(\{Q_i\}_{i\in[d]}), P_{i'}\right\rangle\right| \leq \alpha\right\} \tag{37}$$

$$= \prod_{i'\in[m]\setminus I}\mathbb{P}\left\{\left|\left\langle\theta(\{Q_i\}_{i\in[d]}), P_{i'}\right\rangle\right| \leq \alpha\right\} \tag{38}$$

$$= (\mathbb{P}\{|\langle\theta, P_1\rangle| \leq \alpha\})^{m-d}, \tag{39}$$

for arbitrary $\theta \in \mathbb{S}^{d-1}$ due to the independence of events $\left\{\left|\left\langle\theta(\{Q_i\}_{i\in[d]}), P_{i'}\right\rangle\right| \leq \alpha\right\}$. Let $S_{d-1}(r) = \frac{2\pi^{\frac{d}{2}}r^{d-1}}{\Gamma(\frac{d}{2})}$ be the area of the surface of $d$-dimensional sphere with radius $r$. It is known that

$$\mathbb{P}\{\langle\theta, P_1\rangle > \alpha\} = \frac{\int_\alpha^1 S_{d-2}(\sqrt{1-x^2})\frac{dx}{\sqrt{1-x^2}}}{S_{d-1}(1)} \tag{40}$$

$$= \frac{S_{d-2}(1)}{S_{d-1}(1)}\int_\alpha^1\left(\sqrt{1-x^2}\right)^{d-3}dx. \tag{41}$$

$$= \frac{\Gamma(\frac{d}{2})}{\sqrt{\pi}\Gamma(\frac{d-1}{2})}\int_\alpha^1\left(1-x^2\right)^{\frac{d-3}{2}}dx. \tag{42}$$

Therefore, for $0 \leq \alpha \leq \frac{1}{4}$,

$$\mathbb{P}\left\{|\langle\theta, P_1\rangle| > \alpha\right\} = \frac{2\Gamma(\frac{d}{2})}{\sqrt{\pi}\Gamma(\frac{d-1}{2})} \int_\alpha^1 \left(1 - x^2\right)^{\frac{d-3}{2}} dx \tag{43}$$

$$\geq \frac{2\Gamma(\frac{d}{2})}{\sqrt{\pi}\Gamma(\frac{d-1}{2})} \int_\alpha^{2\alpha} \left(1 - x^2\right)^{\frac{d-3}{2}} dx \tag{44}$$

$$\geq \frac{2\Gamma(\frac{d}{2})}{\sqrt{\pi}\Gamma(\frac{d-1}{2})} \alpha \left(1 - 4\alpha^2\right)^{\frac{d-3}{2}} \tag{45}$$

$$\geq \frac{1}{\pi} \left(1 - 4\alpha^2\right)^{\frac{d-3}{2}} \tag{46}$$

$$\geq \frac{1}{\pi} \exp\left(-4d\alpha^2\right). \tag{47}$$

Therefore, setting $\rho = \frac{m}{d}$, we see

$$\mathbb{P}\left\{\check{r}(K) > \alpha\right\} \leq \mathbb{P} \bigcup_{\{Q_i\}_{i\in[d]}} \bigcap_{i\in[m]} \left\{\left|\left\langle\theta(\{Q_i\}_{i\in[d]}), P_i\right\rangle\right| \leq \alpha\right\} \tag{48}$$

$$\leq \sum_{\{Q_i\}_{i\in[d]}} \mathbb{P} \bigcap_{i\in[m]} \left\{\left|\left\langle\theta(\{Q_i\}_{i\in[d]}), P_i\right\rangle\right| \leq \alpha\right\} \tag{49}$$

$$\leq \binom{2m}{d} \left(1 - \frac{1}{\pi}\exp(-4\alpha^2 d)\right)^{m-d} \tag{50}$$

$$\leq \left(\frac{2me}{d}\right)^d \left(\exp\left(-\frac{1}{\pi}\exp(-4\alpha^2 d)\right)\right)^{m-d} \tag{51}$$

$$= \exp\left(d\left(\log(2\rho e) - \frac{(\rho-1)}{\pi}\exp\left(-4\alpha^2 d\right)\right)\right). \tag{52}$$

Let $\alpha = \sqrt{\frac{s\log\rho}{d}}$. Then, $\alpha < \frac{1}{4}$ implies $\rho < e^{\frac{d}{16s}}$. Then, (52) is rewritten as follows:

$$\mathbb{P}\left\{\check{r}(K) > \sqrt{\frac{s\log\rho}{d}}\right\} \leq \exp\left(d\left(\log(2\rho e) - \frac{\rho-1}{\pi}\rho^{-4s}\right)\right). \tag{53}$$

Since it holds that

$$\log(2\rho e) - \frac{\rho-1}{\pi}\rho^{-4s} \leq -\sqrt{\rho} \Leftrightarrow \log(2\rho e) + \frac{1}{\pi}\rho^{-4s} + \sqrt{\rho} \leq \rho^{1-4s}, \tag{54}$$

when $s < \frac{1}{8}$ there exists $\rho$ that satisfy this inequality. In case $s = \frac{1}{64}$ this inquality satisfies if $e^4 < \rho$. This implies our main claim

$$\mathbb{P}\left\{\check{r}(K) > \frac{1}{8}\sqrt{\frac{\log\rho}{d}}\right\} \leq \exp\left(-d\sqrt{\rho}\right), \tag{55}$$

holds as long as $e^4 \leq \rho \leq e^{4d}$. $\qquad\square$

Now we prove Theorem 2.

**Proof of Theorem 2**

*Proof.* Condition $a \leq \lambda$ implies $\mu \leq \lambda$ from $\mu \leq \min_{\ell\neq\ell'} \text{aff}(\mathcal{S}_\ell, \mathcal{S}_{\ell'})$. From Lemma 4, we can see that condition $\lambda < \frac{1}{8}\sqrt{\frac{\log\rho}{d}}$ implies $\lambda < \check{r}$ with probability greater than $1 - L\exp(-d\sqrt{\rho})$ when $4 < \log\rho < 4d$ is satisfied by the union bound of the probability. Therefore, we obtain the conclusion from the result of Theorem 1. $\qquad\square$

## Appendix D: Random sampling case

In this Appendix, we show that randomly sampled subsamples also enjoy SDP similar to Theorem 1 and Theorem 2 for selective subsamples. This broadens the applicability of our analysis to other works such as SSSC[4, 5]. We first formally state the proposition as follows.

**Proposition 3** (Compatible proposition to Theorem 1). *Assume that data $\mathbf{X} \in \mathbb{R}^{M \times N}$ with normalized columns and subspaces $\{\mathcal{S}_\ell\}_{\ell \in [L]}$ are given. We define $\ell(i)$ so that the subspace corresponding to $i$-th data is $\mathcal{S}_{\ell(i)}$ and $S_\ell = \{i \in [N] | \ell(i) = \ell\}$. Assume that $|S_\ell| = N/L$ and $\dim \mathcal{S}_\ell = d$, for all $\ell \in [L]$. $\mathbf{X}[S_\ell]$ denotes the subset which corresponds to data in $S_\ell$. We define*

$$\check{r} = \min_\ell \check{r}(\mathbf{X}[S_\ell]), \; \mu = \max_{\ell \neq \ell'} \mu\left(\mathbf{X}[S_\ell], \mathbf{X}[S_{\ell'}]\right), \tag{56}$$

*and assume $\lambda$ satisfies*

$$0 < \mu \leq \lambda < \check{r}. \tag{57}$$

*Let $S_T'$ be a set of the indices of randomly chosen subsamples with cardinality $T$ among $[N]$, where*

$$T \geq 2\left(1 + \frac{L}{d}\left(\log(2L\delta^{-1})\right)\right)dL \tag{58}$$

*Then, consider the following problem:*

$$\underset{(C_{ji})_{j \in [N]} \in \mathbb{R}^N}{\text{minimize}} \frac{1}{2} \left\| \mathbf{x}_i - \sum_{j \in [N]} C_{ji}\mathbf{x}_j \right\|_2^2 + \lambda \sum_{j \in [N]} |C_{ji}|$$

$$\text{subject to } C_{ji} = 0, \forall j \in \{i\} \cup ([N] \setminus S_T'), \tag{59}$$

*Then, the matrix formed by the solutions $\mathbf{C}^* = (C_{ji}^*)_{ji}$ satisfies SDP with the probability at least $1 - \delta$.*

*Proof.* From Lemma 3, we can see that if $T \geq 2\left(1 + \frac{L}{d}\left(\log(2L\delta^{-1})\right)\right)dL$, the following holds:

$$\mathbb{P}\left\{\min_\ell |S_\ell \cap S_T'| \geq d\right\} \geq 1 - \delta. \tag{60}$$

Then, we check SDP of the solution similarly as in the proof of Theorem 1. $\qquad\square$

The compatible proposition to Theorem 2 can be proven similarly as the argument above.

**Proposition 4** (Compatible proposition to Theorem 2). *Assume that data $\mathbf{X} \in \mathbb{R}^{M \times N}$ is drawn from semi-random model in which subspaces $\{\mathcal{S}_\ell\}_{\ell \in [L]}$ are given. We define*

$$\rho = \frac{N}{dL}, \; a = \min_{\ell \neq \ell'} \text{aff}(\mathcal{S}_\ell, \mathcal{S}_{\ell'}). \tag{61}$$

*Assume that $\lambda$ and $\rho$ satisfies*

$$4 < \log \rho < 4d, \; a \leq \lambda < \frac{1}{8}\sqrt{\frac{\log \rho}{d}}. \tag{62}$$

*Let $S_T'$ be a set of the indices of randomly chosen subsamples with cardinality $T$ among $[N]$, where*

$$T \geq 2\left(1 + \frac{L}{d}\left(\log(2L\delta^{-1})\right)\right)dL. \tag{63}$$

*Then consider the following problem:*

$$\underset{(C_{ji})_{j \in [N]} \in \mathbb{R}^N}{\text{minimize}} \frac{1}{2} \left\| \mathbf{x}_i - \sum_{j \in [N]} C_{ji}\mathbf{x}_j \right\|_2^2 + \lambda \sum_{j \in [N]} |C_{ji}|$$

$$\text{subject to } C_{ji} = 0, \forall j \in \{i\} \cup ([N] \setminus S_T'), \tag{64}$$

*Then, the matrix formed by the solutions $\mathbf{C}^* = (C_{ji}^*)_{ji}$ satisfies SDP with the probability at least $1 - \delta - L\exp(-d\sqrt{\rho})$.*

*Proof.* By the same argument of Theorem 2, conditions

$$4 < \log \rho < 4d, \ a \le \lambda < \frac{1}{8}\sqrt{\frac{\log \rho}{d}}, \tag{65}$$

imply that

$$\mathbb{P}\{\mu \le \lambda < \check{r}\} \ge 1 - L\exp(-d\sqrt{\rho}). \tag{66}$$

Therefore, Proposition 1 implies Proposition 2 by the union bound of the probability. □

## Appendix E: Experimental setup

In all experiments, we carefully tuned the parameters of all algorithms. For S$^5$C and SSSC [4, 5] we tuned the parameter $\lambda$ in $[2^{-1}, 2^{-10}]$ with exponential step 2. The batch size $B$ in our method was set to 1. For Nyström [6] and AKK [7] parameter $\gamma$ was tested in $[2^{-13}, 2^3]$ with exponential step 4. Parameter $\alpha$ in SSC [8] was chosen in set $\{5, 10, 20, 50, 80, 100, 200, 500, 800\}$. To make EnSC-ORGEN [9] comparable to other SSC algorithms, we set $\ell_2$ norm regularizer to small value (0.001) and optimize sparsity regularizer in the same range as ours. We denote this algorithm by SSC-ORGEN. For SSC-OMP [10] we optimize number of neighbors in range $[3, 17]$ with step 2. For all compared methods, we use the source codes provided by the authors. If the authors provide best parameters for a dataset, we report result obtained with these parameters. To ensure a fair comparison, we set number of subsamples to the value of 20 times number of clusters for all methods having that parameter. All the experiments were executed on Linux (CentOS 6.4) machines with 96 GB memory and Intel Xeon X5690 CPU (3.47 GHz). Execution codes of all algorithms are implemented in MATLAB and excuted by MATLAB 2016b.

Table 1 shows summary of benchmark datasets.

Table 1: Datasets summary: number of data points ($N$), number of dimensions ($M$) and number of clusters ($L$).

| **Dataset** | $N$ | $M$ | $L$ |
|---|---|---|---|
| Yale B | 2432 | 2016 | 38 |
| COIL-100 | 7200 | 1024 | 100 |
| Letter-rec | 20000 | 16 | 26 |
| CIFAR-10 | 60000 | 3072 | 10 |
| MNIST | 70000 | 784 | 10 |
| Devanagari | 92000 | 1024 | 46 |