[Reviews · NeurIPS 2019]

Reviewer 1



This is a well-written original paper. The key intuition of the proposed algorithm is neat. The effectiveness of the algorithm is well supported by theoretical guarantees and experimental results. I think this paper's contribution is sufficiently significant for NeurIPS publication. One main question is that I wonder how this algorithm performs in noisy subspace clustering. I think that limiting the support set will make the algorithm performing worse in noisy setting. Some experiments with synthetic data will help understanding this. [Minor comments/Typos] (3) : subject to C_{ii} = 0, ==> subject to C_{i' i} = 0, (9) : min aff ==> max aff? Shouldn't max be upper bounded in the sufficient condition?

Reviewer 2



The proposed algorithm is well motivated and provided significant improvement over the original SSC in terms of computational cost. The approximation has been well justified by performance guarantees that hold with high probability. I think the contributions are significant. On the other hand, there are several questions and major issues particularly with the presentation. 1. The algorithm is not clearly presented in the main manuscript. (It was better described in an appendix in the supplementary material.) To understand the implication of the main theoretic results, it will be important to see the key structures of the algorithm. So I suggest to elaborate on Section 3.1 in this perspective. 2. Continued from the previous item, due to some missing critical details about the algorithm, the statement of Theorem 1 is not very clear. For example, there is no randomness in the model. The only source of randomness is random selection per iteration within the algorithm. Without clearly explaining this aspect, one may wonder what the probability means in the statement of the theorem. 3. Another issue with the statement of the main theorems is the interpretation of the result. T is rather the size of selection. The number of iteration has to do with how the algorithm finds S. So its meaning as the cardinality of S looks more important. 4. It would be interesting to see how the main results reduces to the analogous results by the original SSC. Does it reproduce the original result as a special case? 5. How can the result be compared to fully stochastic setup where new selection is given not by an optimization of a given large set but by random sampling of the union of subspaces? What would be a gain from "optimizing" from a given set rather than purely random increment? 6. As the authors discussed, the original SSC formulation decouples over columns and can be implemented as a set of Lasso problems in a distributed way. If one uses a cluster to solve the problem, the implementation can be parallelized. For example, each lasso can be solved by FISTA or ADMM. If this is correct, can one still suffer from failures in Table 1 for the original SSC? 7. There were a few critical typos. For example equation (3) and line 193 (SEP?) 8. It might be useful to provide a geometric interpretation of the affinity in Definition 6 in terms of principal angles between subspaces.

Reviewer 3



This paper proposes an incremental algorithm based on a subgradient approximation to select a subsample of the data to improve the computational complexity of Sparse Subspace Clustering (SSC). The authors show that under standard coherence conditions, this subsampling strategy (S5C) will in fact recover the true underlying union of subspaces in the data, thus improving the computational complexity of SSC without compromising performance. In addition, the paper gives several experiments to show the effectiveness of their approach. One of my main concerns is whether the paper is novel/significant enough for NeurIPS. I am also curious as to the choice of datasets for experiments, some of which are not related to/justified for subspace clustering. I am also a bit confused. If S5C is indeed running SSC on a subsample of the data, shouldn't SSC (and other variants that exploit the entire sample) do better than S5C (precisely because they are exploiting the entire sample)? It would be interesting to understand why S5C seems to be performing better. I initially thought it was because the parameters were only fine tuned for S5C, but after carefully reading Appendix E in the supplementary material, it seems like the authors did a very thorough and fair job at fine tuning all other algorithms as well. Hence I am curious as to what is the explanation behind this performance improvement -- which could be very interesting; maybe somehow this sub sampling is helping in some other way, for example ignoring outliers, or something like that? Finally, I think such a thorough review/discussion of SSC, spectral clustering, and some technical definitions could be shrunk considerably. ---- After Rebuttal ---- The authors have addressed my concerns, so I am changing my score to accept.

[Author Response · NeurIPS 2019]

We thank the reviewers for their valuable comments and recommendations for the improvement. Overall, it seems that the reviewers R1 and R2 found our contributions significant, but had questions about presentation of our theoretical results (R2), comparison of our theoretical results to the existing methods and random sampling (R1, R2), and practical comparison to SSSC (R1). R3 asked about significance and choice of datasets. R3 also suggested to shrunk review of SSC; we will do so and use space to address concerns of R2 by clarifying theoretical results.

**Significance (R3).** We would like to point out that our contributions are twofold. We present first linear time algorithm which directly solves SSC objective function and show it outperforms other large-scale SSC motivated algorithms. Besides algorithmic contribution, we provide novel theoretical result of SSC for limited number of subsamples. Our analysis gives theoretical guarantees needed for the success of SSC in the setting of limited number of subsamples.

**Clarification of theoretical result (R1, R2).** We highly appreciate that the reviewers pointed out to improve presentation of theoretical analysis. Due to the space constraints, pseudocode of the algorithm was included in the Appendix, but we will correct this and present key steps in the paper. We agree with R2 that this will help in understanding theoretical results. R2 correctly pointed out that the randomness in the model comes from random selection per iteration in the algorithm. We will clarify this point and relate to the probability in the statement of the theorems. Furthermore, number of iterations $T$ is directly and linearly connected to the runtime of the algorithm. At the same time, $T$ has an interpretation as an upperbound on the cardinality of $S$. We will explain this in the final version. Affinity in Definition 6 is defined in terms of principal angles in references [12, 13] in the paper; we will cite [12, 13] for alternative definition.

**Compatibility to the existing results and stochastic variant (R1, R2).** The setting of Theorem 1 is comparable to the setting considered in the analysis of SSC-OMP [21], in which authors assume that data is noiseless and each pair of subspaces is arbitrarily intersected. Authors further assume that all data is used as a subsample so that it coincides with the original SSC, whereas our analysis first succeeded to show that smaller number of subsamples is sufficient to guarantee SDP. The setting of Theorem 2 is adopted from the setting of noisy SSC [12], where data is randomly drawn from each subspace. Our analysis relies on the notion of persistent inradius, which is a measure originally introduced in our work. When inradius and persistent inradius coincide, the SSC results can be reproduced as a special case. R1 was interested about noisy SSC setting. Our theoretical result does not assume noise, but empirically we perform well in the noisy setting of real-word data. We will add part about the compatibility to the existing SSC theoretical results.

R2 asked about the comparison to the fully stochastic variant. Our theoretical analysis shows that, in the case of random sampling, $T$ subsamples are needed to satisfy SDP, while S$^5$C needs a smaller number of subsamples to satisfy SDP. We will clarify this in the revised version.

**Table 1:** Relation to the existing theoretical analyses

|  | subsample | noise | data model | measure for subspaces | condition on data |
|---|---|---|---|---|---|
| Theorem 2 in [21] | no | no | deterministic | incoherence | large inradius |
| Theorem 2.8 in [12] | no | yes | semi-random | affinity | large number of data |
| Our Theorem 1 | yes | no | deterministic | incoherence | large persistent inradius |
| Our Theorem 2 | yes | no | semi-random | affinity | large number of data |

**Choice of datasets (R3).** Yale B, Hopkins 155 and MNIST are the most benchmarked subspace clustering datasets. We did not compare performance only on Hopkins 155 dataset, but per reviewer's question we now include Hopkins dataset. We report average clustering error across 155 sequences (Table 2), after carefully tuning parameters of all algorithms. Results show that S$^5$C significantly outperforms all other large-scale methods. We will include these results in the final version. Among other datasets, only Devanagari has not been previously used for subspace clustering. However, the use of this dataset is justified since it is a large-scale dataset similar to MNIST, but the problem is even harder: instead of handwritten digits the task is to recognize handwritten letters.

**Table 2:** Average clustering error across 155 sequences of Hopkins 155

| Dataset | Nyström | AKK | SSC | SSC-OMP | SSC-ORGEN | SSSC | S$^5$C |
|---|---|---|---|---|---|---|---|
| Hopkins 155 | 21.8 | 20.6 | **4.1** | 23.0 | 20.5 | 21.1 | **15.8** |

**Experimental comparison to baselines (R1, R2, R3).** R3 expressed concern that we report better performance than the methods that exploit all samples. Reviewer might be confused by the names of the methods (SSC-OMP, SSSC), but the only method that exploits all samples is SSC and it performs better than any other method (including ours). We will change name of SSC-ADMM to SSC to avoid ambiguity.

R1 raises a point about the practical benefits of our method compared to SSSC. Our method has significantly better performance than SSSC, achieving even 14.5% better average performance across all datasets (Table 1 in the paper). To demonstrate additional benefits, we show parameter sensitivity of SSSC and S$^5$C on Devanagari dataset (Figure 1). S$^5$C outperforms SSSC for all values of sparsity regularizer $\lambda$ and number of subsamples. Furthermore, when the number of subsamples is increased S$^5$C expectedly achieves lower clustering error, while SSSC does not exhibit such behavior. Similar behavior can be observed on other datasets.

Finally, R2 was interested whether distributed computing of SSC can prevent the problem. Since the original SSC suffers from $O(N^3)$ time complexity, the effect of distributed computing is limited. On the other hand, since our algorithm is $O(N)$, we can deal with linearly larger number of datapoints as available computational resources increases.

**Figure 1:** Clustering error of S$^5$C and SSSC

[Meta-Review · NeurIPS 2019]

This paper describes a variant of sparse subspace clustering in which scalability issues are addressed by running subspace clustering on subsets of the data at a time. Previous work addressed this for random subsets, but this paper considers what amounts to a more targeted sampling approach that leads to non-trivial empirical performance improvements on real datasets and accompanying theory.